# Aggressive Intravenous Hydration and a Defined Plant-Based Diet Safely and Effectively Treated Type 5 Cardiorenal Syndrome with Stage E Heart Failure-Related Cardiogenic Shock: A Case Report

**DOI:** 10.3390/reports7040094

**Published:** 2024-11-08

**Authors:** Baxter Delworth Montgomery, Camille V. Owens, Rami Salim Najjar, Mawadda Saad

**Affiliations:** 1Montgomery Heart & Wellness Center, 10480 S Main St., Houston, TX 77025, USA; cowens@montgomeryheart.com (C.V.O.); msaad@montgomeryheart.com (M.S.); 2Institute for Biomedical Sciences, Georgia State University, 161 Jesse Hill Jr Dr SE, Atlanta, GA 30303, USA; rnajjar1@gsu.edu

**Keywords:** kidney disease, heart failure, plant-based diet, polypharmacy, hydration

## Abstract

**Background and Clinical Significance**: Heart failure and kidney diseases often coexist and are difficult to clinically manage. Dysfunction in either organ exacerbates dysfunction in the other, potentially leading to cardiorenal syndrome (CRS). CRS has five different subtypes, with CRS type 5 being the most problematic given that it consists of an acute insult superimposed upon chronic CRS. Additionally, type 5 CRS can be complicated by heart failure-related cardiogenic shock (HF-CS), which is associated with increased hospitalizations and has a high 1-year mortality rate. The standard treatment for patients with HF-CS consists of guideline-directed medical therapy for heart failure with reduced ejection fraction (HFrEF) as tolerated, along with inotropic therapies and surgical mechanical left ventricular (LV) support, guided by invasive hemodynamic monitoring. **Case Presentation**: This case study reports the presentation of a 57-year-old man who presented with type 5 CRS who rapidly decompensated to stage E HF-CS and was effectively and safely treated with aggressive intravenous hydration, a defined plant-based diet (DPBD), and reduction of guideline-directed prescription medications without invasive hemodynamic monitoring. **Conclusions**: Hydration, a DPBD, and a reduction in medication burden may be effective in CRS. Pilot studies are warranted to evaluate the efficacy of this intervention in CRS in a larger cohort.

## 1. Introduction and Clinical Significance

Cardiac and renal diseases comprise the top 10 leading causes of death in the United States [1]. Up to 63% of patients with heart failure have some form of cardiorenal syndrome (CRS) [2]. CRS is a complex multiorgan clinical condition in which acute or chronic cardiac or renal dysfunction accelerates the decline in the other organ [3]. The medication burden is typically high in patients with CRS, and diuretics are often used to reduce edema caused by either heart failure, renal failure, or both. As a result of this, the volume of the intravascular space in these patients may be chronically and sub-clinically depleted.

Type 5 CRS is characterized by chronic cardiac and renal dysfunction exacerbated by a systemic metabolic insult, such as sepsis, drug toxicity, or systemic inflammation. Such a systemic insult can result in simultaneous worsening of cardiac and renal function. This results in poor cardiac output, which leads to poor glomerular filtration due to poor perfusion of the kidney, causing hypoxia and kidney damage [3]. Worsening kidney function leads to metabolic derangements such as metabolic acidosis and electrolyte abnormalities. These metabolic abnormalities lead to the acceleration of heart failure. As a result, patients with type 5 CRS are at a greater risk of decompensating into HF-CS due to the systemic metabolic insults on their chronically impaired cardiac and renal function.

The relatively high likelihood of chronic intravascular volume depletion in the setting of CRS can create special diagnostic and therapeutic challenges when these patients decompensate to HF-CS. For example, invasive hemodynamic monitoring may be misleading, as elevated pulmonary capillary wedge pressure (PCWP) and right atrial (RA) pressure measurements could be elevated due to either increased volume or poor LV or RV compliance, respectively [4]. Additionally, these patients would likely not tolerate inotropic therapies due to a reduction in perfusion in the setting of severe volume depletion. Moreover, patients with stage E HF-CS may have severe metabolic acidosis as a result of severe hypoperfusion [5]. This can result in the worsening of underlying cellular and tissue-level inflammation and oxidative stress [6].

Alternative therapies in the treatment of CRS are scant in the literature. However, we currently present a case study of a 57-year-old man who presented with type 5 CRS in the setting of medication-induced severe volume depletion and acidosis. Upon initial presentation, the patient was not severely acidotic or in HF-CS. However, within 7 days of initial outpatient presentation, he decompensated to HF-CS with associated severe metabolic acidosis. After hospitalization, he was safely and effectively treated with a defined plant-based diet (DPBD), aggressive intravenous hydration, and a reduction of guideline-directed prescription medications.

## 2. Case Presentation

### 2.1. Patient Presentation

A 57-year-old African American male with congestive heart failure, an implantable defibrillator, type 2 diabetes, stroke, coronary artery disease, sleep apnea, hypertension, chronic kidney disease of an unknown stage, and gout presented with worsening shortness of breath and chest pain on 14 May 2020. He was told by a previous healthcare provider that he needed a heart transplant roughly two years prior. He presented with symptoms of severe substernal chest tightness at least once per week, which began one month prior to his presentation. This chest discomfort occurred at rest and worsened with exertion. He also reported shortness of breath and fatigue with mild activities, in addition to abdominal distention and bloating. The patient was on 11 prescription medications, which included allopurinol, Vitamin D2, carvedilol, Entresto™, torsemide, Klor-Con™, Tradjenta™, Humalog™, Lantus™, atorvastatin, and Eliquis™ (Appendix A).

The baseline physical examination revealed a normal heart rate and rhythm, without audible murmur. His lungs were clear to auscultation without rales, rhonchi, or wheezing. His peripheral pulses were palpable with a normal amplitude, and his lower extremities were without edema. However, his abdomen was distended, with dullness to percussion. His initial vital signs revealed a blood pressure of 96/48 mmHg, a heart rate of 76 bpm, a BMI of 32 kg/m^2^, and an oxygen level of 98% on room air. Cardiovascular testing was performed at the outpatient clinic. A resting EKG showed poor r wave progression, left axis deviation, frequent premature ventricular complexes (PVCs), and a left bundle-like intraventricular conduction delay (Appendix A). A treadmill stress test utilizing the Bruce protocol showed a reduced functional capacity with an exercise time of 1 min and 4 s and frequent PVCs. An echocardiogram revealed a left ventricular ejection fraction (LVEF) of 20%, moderate left ventricular dilation, and segmental wall motion abnormalities (Appendix A). His creatinine levels were 5.08 mg/dL, his eGFR was 14 mL/min/1.73 m^2^, his carbon dioxide levels were 18 mmol/L, and his urea nitrogen levels were 176 mg/dL. Potassium was normal.

### 2.2. Intervention

The patient initially declined hospitalization and hence was started on a DPBD, along with a reduction in his baseline diuretic medication dosage by our clinical team [7]. This diet consisted only of minimally processed vegetables, fruits, and seeds. Beverages included water, cold-pressed fruit and vegetable juices, and raw, nondairy smoothies. There was no fluid volume restriction. Foods of animal origin and free oils were not permitted. Based on a nutritional analysis of this identical dietary strategy we published prior, fruit and vegetable intake was increased to ~12 and 16 servings/day, respectively [8]. An additional nutrient analysis is detailed in our prior work, which includes macronutrient and micronutrient intake [7]. Other medication changes were made based on the patient’s clinical response to the dietary intervention and repeat physical assessments and lab analysis. The decision-making process for medication changes was previously described by our group [7].

### 2.3. Clinical Course

After four days on the DPBD, the patient reported by phone that his angina and difficulty breathing had resolved. He reported more frequent bowel movements and an increase in his energy level. Entresto™, Klor-Con™, Tradjenta™, insulin, allopurinol, and atorvastatin were discontinued at this time. Torsemide was reduced by half. He was seen in the clinic on day 7 of the DPBD and reported black stool, a reduced appetite, lightheadedness with ambulation, and a recurrence of shortness of breath. On repeat physical examination, he was hypotensive, with a blood pressure of 69/41 mmHg. His oxygen was normal, and his heart rate averaged 86 bpm during his visit. He was given 2000 mL of 0.9% sodium chloride solution intravenously due to signs and symptoms of intravascular volume depletion. As a result, his blood pressure increased to 100/64 mmHg. Eliquis was discontinued due to concerns of gastrointestinal bleeding. Torsemide was discontinued completely. Labs were repeated and sent off stat. He was seen again in the clinic on day 8, and a review of the repeat labs revealed worsening renal function and metabolic acidosis with creatinine levels of 12.10 mg/dL, an eGFR of 4 mL/min/1.73 m^2^, and carbon dioxide levels of 11 mmol/L (Table 1). The patient was immediately hospitalized after this evaluation.

Upon admission, the patient was started on milrinone for inotropic support, but it was discontinued due to worsening tachycardia. A cardiac nuclear stress test was performed that showed a medium-sized, moderate-intensity anterior and apical non-transmural scar, with an ejection fraction of less than 20%, indicating both ischemic and non-ischemic cardiomyopathy. The patient was also found to be severely acidotic on admission with an initial arterial blood gas pH of 7.12. He was then hydrated with an intravenous (IV) solution of 75 mEq of 8.4% sodium bicarbonate in 1000 mL of 0.45% sodium chloride solution for treatment of his severe volume depletion and acidosis. This was administered at a rate of 100 mL/h. The patient received 6 L of intravenous fluids during his hospitalization (the first 2 L were infused before the initial arterial blood gas measurement). Early on, his pH, base excess, and PaO_2_ levels were assessed multiple times a day using arterial blood gas measurements until normalization (Figure 1). His electrolytes and other metabolic parameters were closely monitored also. His lactic acid levels were initially normal and remained so throughout his hospital course. There was a direct correlation of the infusion of fluids with the correction of his acidosis (Figure 1A) and an improvement in his eGFR (Figure 2). Additionally, his oxygen saturation normalized and remained stable above 96% throughout his hospitalization. (Figure 1C). His eGFR gradually increased to 28 mL/min/1.73 m^2^, his creatinine levels decreased to 2.78 mg/dL, and his blood carbon dioxide normalized (Figure 2). Throughout his hospital stay, he continued consuming the DPBD. After discharge, he was on two prescription medications only, which included Vitamin D3 (changed from Vitamin D2) and Bystolic.

Following hospitalization, the patient was no longer experiencing lightheadedness, weakness, chest pain, or shortness of breath. At his first outpatient follow-up visit, a repeat treadmill stress test was performed, this time utilizing the Modified Bruce protocol. The patient exercised for 4 min and 18 s. He continued a DPBD for 13 more days. During this process, his energy level continued to improve, and his abdominal bloating decreased. A second Modified Bruce treadmill stress test was performed, and this time, the patient exercised for 13 min and 3 s. On a follow-up EKG, the previously frequently seen PVCs were no longer present. A lab analysis revealed additional changes in his renal function from the time of hospital discharge. These included a 35% decrease in creatinine (creatinine level of 1.80 mg/dL) and a 68% increase in eGFR (eGFR now 47 mL/min/1.73 m^2^). Overall, his eGFR increased by 236%, and there was an 82% decrease in his medications, from 11 to 2.

## 3. Discussion

The patient in this case report presented with type 5 CRS and decompensated to HF-CS within one week of his presentation. The factors contributing to his heart failure and chronic kidney disease were atherosclerosis, type 2 diabetes, and hypertension. He had been treated with a high dose of torsemide and Entresto™, both of which likely contributed to his volume depletion and renal dysfunction. He was also on nine other medications, which may have also contributed to his cardiac and renal dysfunction. His clinical symptoms of chest pain and shortness of breath could have been due to decompensated heart failure with volume overload or cardiac ischemia due to volume depletion. The presentation of hypotension and metabolic acidosis was an early clue of poor perfusion; however, the extent of this being due to severe volume depletion rather than poor cardiac performance was not immediately clear. However, it was likely that both were contributing factors. Given these factors, it was our opinion that medication-induced volume depletion and associated renal toxicity were the systemic insults that classified him into type 5 CRS.

The patient’s rapid decomposition to HF-CS complicated his clinical management greatly. His intolerance of milrinone made it likely that he would not tolerate any other inotropic drug. The standard recommendations for patients with HF-CS stages A–E range from the continuation of guideline-directed medical therapy and diuresis for hemodynamically stable patients to the use of inotropic drugs and various forms of ventricular mechanical support (venoatrial extracorporeal membrane oxygenation, intra-aortic balloon pumps, left ventricular assist devices, etc.) [5]. Ventricular mechanical support treatments were strongly considered and preliminarily planned for, along with emergency hemodialysis, given the severity of his baseline acidosis and renal failure.

However, given the patient’s severe dehydration and acidosis, we elected to immediately begin with intravenous hydration with sodium bicarbonate added to 0.045% saline solution, with very close monitoring of his pH and base excess. We felt that these metabolic markers would be excellent surrogates for his underlying perfusion. It became evident that his acidosis was clearing within the first several hours of his course of treatment. Over the following 6–18 h, his overall clinical condition showed drastic improvement. This continued for the remainder of his hospitalization while he received aggressive IV hydration and remained on the DPBD.

Treating patients with HF-CS with aggressive IV hydration is somewhat unorthodox. This is because these patients are not thought to tolerate such large volumes of fluid due to either existing volume overload and/or cardiac failure. However, it was our assessment that this patient was suffering from extreme volume depletion that greatly contributed to his cardiac and renal failure. Hence, the aggressive IV hydration was thought to benefit him immediately. In the early hours of his admission, his acidosis was at least in part controlled by hyperventilation, with a drop in his pCO2 correlating with an increase in his pH from 7.12 to 7.18 (Figure 1D). However, over time, his pH continued to correct, with an associated increase in his pCO2, denoting less need for hyperventilation. Additionally, his pH improved (Figure 1A) and his base excess decreased (Figure 1B) during the IV infusion. His oxygenation saturation initially improved and remained over 96% during the duration of the infusion (Figure 1C).

The baseline pathophysiological factors contributing to this patient’s clinical condition likely consisted of systemic inflammation and increased oxidative stress with his chronic CRS. Further, diuretics, while often used to manage fluid overload, can unintentionally worsen renal function. They can lead to excessive fluid and electrolyte losses, potentially causing dehydration and reducing renal perfusion pressure [9]. In this case, medications such as torsemide and Entresto may have resulted in excess intravascular volume depletion. This likely caused his lightheadedness, hypotension, and severe kidney injury, which led to metabolic acidosis. While Entresto™ typically has favorable outcomes in heart failure and renal failure, the natriuretic and diuretic effect of the neprilysin inhibitor in this setting likely had a detrimental effect [5]. The patient was previously chronically dehydrated and ate a standard American diet, which is typically higher in sodium and generally has a lower water content than plant foods. He also was prescribed high doses of torsemide and Entresto. Together, this contributed to a severely reduced volume status and renal perfusion. Furthermore, the patient’s state of polypharmacy likely contributed to his worsening kidney failure and cardiovascular function by means of negative drug interactions, intravascular volume depletion, altered drug metabolism and elimination, and potential organ dysfunction. It was very likely that polypharmacy was associated with kidney dysfunction in this patient [10].

It is important to point out the underlying driving mechanisms of this patient’s baseline progressive clinical decline were biochemical in nature and occurred at the cellular and tissue levels. Chronic hyperglycemia in type 2 diabetes and hypertension can increase oxidative stress at the cellular and tissue levels, driving inflammation and microvascular dysfunction, leading to renal and cardiac injury [4]. Due to his medical therapy, he developed two pre-renal mechanisms (volume depletion and worsening cardiac output) that worsened his renal function further. The further worsening of his cardiac function likely increased RAS activation to compensate for reduced systemic organ perfusion. Increased angiotensin II signaling can increase inflammation and oxidative stress via the angiotensin II type-1 receptor, creating a negative cascade of renal and cardiac dysfunction [11].

The infusion of sodium bicarbonate mixed with the 0.45% saline solution had a direct positive hemodynamic effect by correcting the volume depletion while at the same time correcting the acidosis. It is clear that this regimen likely contributed to his immediate clinical improvement in the initial hours of his hospitalization.

Our choice of using sodium bicarbonate mixed in a 0.45% saline solution is an important one and deserves a brief discussion. The patient in this presentation was both severely volume-depleted and acidotic. An isotonic crystalloid solution is ideal for volume depletion. Normal saline (0.9%) is most commonly available. However, 0.9% saline would not have been ideal for this patient or anyone in a similar condition for two reasons. First, the use of 0.9% saline would have superimposed hyperchloremic metabolic acidosis onto his current metabolic acidosis. This could have led to a greater hemodynamic decline. Secondly, the use of excess 0.9% saline would have blunted the potential triggering of his functional renal reserve mechanism by the sodium. In short, the functional renal reserve consists of a rapid increase in the glomerular filtration rate (GFR) as a result of increased function at the nephron level. This is brought on by a net increase in glomerular flow. Increased renal blood flow and plasma flow are the primary contributors to this increase in flow. Although the mechanism has not been fully elucidated, nitric oxide is thought to play a role. Conversely, increased renal congestion in the distal tubules reduces the GFR because of the adverse mechanical effects of compressing the nephrons with their restrictive encapsulation. Hence, a tubuloglomerular feedback (TGF) mechanism exists that counters the increase in renal flow to prevent this mechanism [12].

Hence, the GFR is increased by the increase in renal flow and decreased by the TGF mechanism. Intravenous volume expansion improves renal flow according to multiple mechanisms, with the major one being an increase in cardiac output (CO). However, an excessive increase in the blood-flow-induced volume to the distal tubules can trigger TGF and hence blunt the increase in the GFR. Additionally, the exposure of the distal tubules to very high levels of chloride exacerbates this effect. Therefore, the use of 0.45% saline with sodium bicarbonate using a carefully controlled infusion rate allowed for the increase in renal flow without an increase in distal tubular congestion. This allowed for maximizing the increase in the GFR using the renal functional reserve mechanism.

Additionally, the DPBD likely played an important synergistic role in his recovery. The DPBD likely contributed to his improved renal and cardiac function mechanistically by reducing oxidative stress and inflammation due to the presence of phytochemicals [13]. Further, metabolic acidosis, a common feature of CKD, can be treated with fruit and vegetable intake due to the reduced renal acid load and alkalinity of plant foods [14]. Indeed, animal-based foods have a much greater renal acid load compared to plant foods, which are alkaline [15]. This is important because the simultaneous removal of animal-based foods from his diet during this treatment phase further enhanced the acid load reduction. Campbell and Liebman reported a similar case study in which a patient with stage 3 CKD improved their eGFR from 45 to 74 mL/min after 4.5 months of following a whole-food plant-based diet. It is noteworthy that the changes observed in this study were more rapid in a more acutely ill patient, although CRS was not present in the aforementioned patient [16].

In the context of discussing the benefits of the DPBD, one should also consider potential adverse effects of this treatment approach in the setting of acutely ill patients. This patient had a subsequent decline in his condition between day 4 and day 7. His initial clinical improvement was very likely due to the effects of the DPBD. The reduction in inflammation, oxidative stress, and vascular resistance can occur withing the first week of this regimen. However, these positive physiological changes when superimposed with medications can potentially be detrimental. A further decrease in blood pressure in the setting of systemic volume depletion could worsen cardiorenal perfusion, resulting in worsening clinical decompensation. This is likely what happened with this patient. It has been our experience that careful monitoring of these patients with aggressive medication weaning is important for their management.

Lastly, the third important aspect of this patient’s treatment consisted of medication weaning. The discontinuation of torsemide likely slowed the progression of CRS. After the complete removal of torsemide and most other medications, as well as the administration of IV hydration in the hospital setting, his renal perfusion improved from an eGFR of 4 to 28 mL/min/1.73 m^2^ and then later to 47 mL/min/1.73 m^2^.

The current standard clinical recommendations for the management of patients with HFrEF in general and HF-CS specifically consist solely of the use of guideline-directed medical therapy as tolerated with the addition of inotropic drugs and/or mechanical LV assist therapies. There are no control studies showing the overall benefit of these therapies for HF-CS patients. This case report and another published case series [17] raise the question of the potential benefit of aggressive metabolic therapies such as hydration and defined nutritional interventions as a potential adjunct or independent therapy in these patients.

More work with controlled studies needs to be undertaken to delineate clearer guidelines for patient selection and detailed treatment implementation. However, HF patients treated with multiple medications, including diuretics, should be evaluated for volume depletion. This population of patients could benefit from careful medication weaning and volume expansion.

## 4. Conclusions

We show that a patient with type 5 CRS with subsequent decompensation to HF-CS could be safely and effectively treated with increased hydration; a reduction in medications, especially diuretics; and the consumption of a defined plant-based diet. Additionally, this case report highlights the potential importance of reducing polypharmacy, improving fluid intake, and improving dietary quality in patients with cardiac and renal dysfunction. The beneficial effects observed are likely synergistic and not exclusively due to any one interventional change. A pilot clinical trial is warranted to explore this intervention further in this patient population.

## Figures and Tables

**Figure 1 reports-07-00094-f001:**
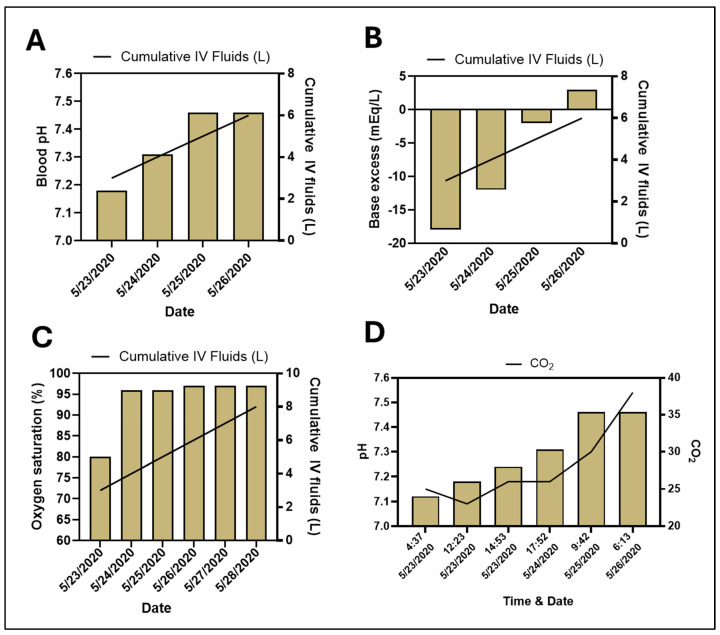
Correlation with (**A**–**C**) IV fluid intake, (**D**) CO_2_, and various metabolic parameters.

**Figure 2 reports-07-00094-f002:**
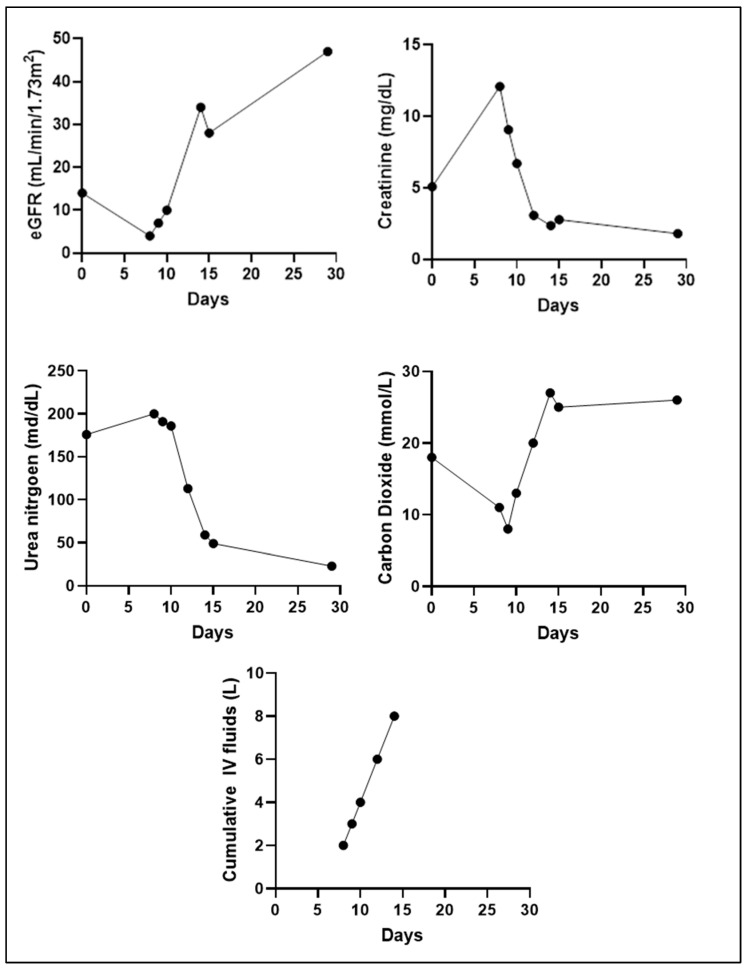
Changes in kidney function and IV fluids during the patient’s clinical course.

**Table 1 reports-07-00094-t001:** Descriptive changes in kidney function during the patient’s clinical course.

Date	eGFR (mL/min/1.73 m^2^)	Creatinine (md/dL)	Urea Nitrogen (mg/dL)	Carbon Dioxide (mmol/L)	Description
14 May 2020	14	5.08	176	18	First office visit. Started the DPBD.
22 May 2020	4	12.1	200	11	First lab analysis after receiving 2 L of IV 0.9% sodium chloride. Up to this point, all medications except Vitamin D2 were discontinued.
23 May 2020	7	9.07	191	8	During hospitalization, after the administration of 75 mEq of 8.4% sodium bicarbonate in 1000 mL of 0.45% sodium chloride.
24 May 2020	10	6.7	186	13	During hospitalization, after continous infusion of 75 mEq of 8.4% sodium bicarbonate in 1000 mL of 0.45% sodium chloride.
25 May 2020	25	3.08	113	20	During hospitalization.
28 May 2020	34	2.37	59	27	Discharged from hospital.
29 May 2020	28	2.78	49	25	First lab repeat after discharge.
11 June 2020	47	1.8	23	26	Last office visit.

## Data Availability

The original data presented in this study are available on reasonable request from the corresponding author. The data are not publicly available due to privacy concerns.

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
