# Peer review of "Aggressive Intravenous Hydration and a Defined Plant-Based Diet Safely and Effectively Treated Type 5 Cardiorenal Syndrome with Stage E Heart Failure-Related Cardiogenic Shock: A Case Report"

_reports, 2024, doi:10.3390/reports7040094_

Round 1
Reviewer 1 Report
Comments and Suggestions for Authors
I’ll provide a review report illustrating strengths and weaknesses according to my point of view:
Introduction
Strengths
- Effectively establishes the clinical significance of cardiorenal syndrome (CRS) and its impact on public health
- Clearly defines Type 5 CRS and explains its pathophysiology
- Provides relevant context about the challenges of treating patients with this condition
- Successfully builds the case for why this particular case study is worth reporting
Weaknesses
- Could have included more recent epidemiological data about CRS prevalence
- Limited discussion of existing alternative treatment approaches
- Could have provided more context about why traditional treatments often fail
Case Presentation
Strengths
- Detailed patient history and initial presentation
- Clear documentation of vital signs and diagnostic test results
- Well-organized chronological presentation of the case
- Comprehensive documentation of medication changes
- Excellent presentation of laboratory data through tables and figures
- Clear description of the plant-based diet intervention
Weaknesses
- No clear explanation of why specific laboratory tests were chosen over others
- Limited description of the decision-making process for medication discontinuation
- Lack of standardized criteria for assessing patient improvement
- No mention of potential risks or contraindications of the aggressive hydration approach
- Missing details about the specific nutritional composition of the plant-based diet
Discussion
Strengths
- Thorough analysis of the pathophysiological mechanisms
- Clear explanation of how each intervention (hydration, diet, medication reduction) contributed to improvement
- Good integration of existing literature to support the findings
- Balanced consideration of multiple factors contributing to patient improvement
- Appropriate acknowledgment of the unorthodox nature of the treatment approach
Weaknesses
- Limited discussion of potential risks or complications of the treatment approach
- Could have included more comparison with traditional treatment outcomes
- No discussion of potential confounding factors
- Limited exploration of why this approach might not work for all patients
- Could have provided more specific guidelines for implementing this approach in clinical practice
Conclusions
Strengths
- Concise summary of key findings
- Appropriate call for further research through pilot studies
- Recognition of the synergistic nature of the interventions
- Clear practical implications for clinical practice
Weaknesses
- Could have been more specific about limitations of the case report
- Limited discussion of specific patient populations that might benefit most from this approach
- No mention of potential contraindications
- Could have provided more specific recommendations for future research design
Overall Assessment
This case report presents an innovative approach to treating Type 5 CRS using a combination of aggressive hydration, plant-based diet, and medication reduction. The presentation is generally well-organized and supported by detailed clinical data. The main strength lies in the comprehensive documentation of the patient's response to treatment and the thorough discussion of potential mechanisms.
However, the report would benefit from:
1. More detailed discussion of potential risks and limitations
2. Clearer guidelines for implementing this approach in clinical practice
3. More specific recommendations for patient selection
4. Better documentation of the decision-making process for treatment modifications
Despite these limitations, the case report makes a valuable contribution to the literature by presenting an alternative approach to managing complex cardiorenal syndrome cases, particularly when traditional treatments have failed or are contraindicated.
Author Response
Introduction
Weaknesses
- Could have included more recent epidemiological data about CRS prevalence
We have included prevalence data at line 28-29.
- Limited discussion of existing alternative treatment approaches
- Could have provided more context about why traditional treatments often fail
These latter 2 points have been addressed at line 54.
Case Presentation
Weaknesses
- No clear explanation of why specific laboratory tests were chosen over others
We have presented laboratory tests specific to kidney function, and these tests were used to make clinical decisions.
- Limited description of the decision-making process for medication discontinuation
This has been addressed at line 100-101
- Lack of standardized criteria for assessing patient improvement
We have followed standardized clinical guideline recommendations of the American Heart Association in the treatment of CRS, and feel that this does not require elaboration.
- No mention of potential risks or contraindications of the aggressive hydration approach
This has been addressed at lines 234-259
- Missing details about the specific nutritional composition of the plant-based diet
This has been addressed at lines 95-98
Discussion
Weaknesses
- Could have included more comparison with traditional treatment outcomes
There is limited use of alternative treatments for CRS.
- Limited discussion of potential risks or complications of the treatment approach
- No discussion of potential confounding factors
- Limited exploration of why this approach might not work for all patients
- Could have provided more specific guidelines for implementing this approach in clinical practice
We believe that lines 297-301addresses these points.
Conclusions
Weaknesses
- Could have been more specific about limitations of the case report
- Limited discussion of specific patient populations that might benefit most from this approach
- No mention of potential contraindications
- Could have provided more specific recommendations for future research design
We believe that lines 297-301addresses these points.
- More detailed discussion of potential risks and limitations
- Clearer guidelines for implementing this approach in clinical practice
- More specific recommendations for patient selection
- Better documentation of the decision-making process for treatment modifications
Despite these limitations, the case report makes a valuable contribution to the literature by presenting an alternative approach to managing complex cardiorenal syndrome cases, particularly when traditional treatments have failed or are contraindicated.
We have included the following limitations at lines 297-301: More work with controlled studies needs to be done to delineate clearer guidelines for patient selection and detailed treatment implementation. However, HF patients treated with multiple medications including diuretics should be evaluated for volume depletion. This population of patients could benefit from careful medication weaning and volume expansion.
Reviewer 2 Report
Comments and Suggestions for Authors
1. These authors have reported a patient with very complicated medical problems who developed type V cardiorenal syndrome and cardiogenic shock. He was treated with intravenous hydration for presumed intravascular volume contraction, a defined plant-based diet, and adjustment of medications. The authors list 11 prescription medications but do not give the doses. This information may be available in the supplementary table, but I was unable to find the table in the website with the manuscript. Consequently, the dose of these various medications is unknown.
2. Based on the information provided in the case summary, it would appear that the patient did not have any intercurrent event which might explain his sudden change in status. That should probably be specifically stated in the case summary.
3. The authors provide laboratory information regarding renal function. They should include some information about troponin and BNP levels.
4. The patient was placed on a plant-based diet which presumably helped his acid-base status. The authors should provide information about his initial weight at presentation and his weight at his final follow-up. It would also be helpful with the authors provided some basic information about this plant-based diet. For example, how many calories are provided in lipids, protein, and carbohydrates?
Author Response
- These authors have reported a patient with very complicated medical problems who developed type V cardiorenal syndrome and cardiogenic shock. He was treated with intravenous hydration for presumed intravascular volume contraction, a defined plant-based diet, and adjustment of medications. The authors list 11 prescription medications but do not give the doses. This information may be available in the supplementary table, but I was unable to find the table in the website with the manuscript. Consequently, the dose of these various medications is unknown.
Thank you for your valuable feedback throughout your report. The supplementary materials indeed has the full medication list, and this was uploaded prior. For your reference, here is the medication list copied and pasted.
|
05.14.2020 |
06.12.2020 |
1 |
allopurinol 300 mg once a day |
Bystolic 5 mg once a day |
2 |
carvedilol 6.25 mg twice a day |
Vitamin D2 50,000 IU once a week |
3 |
Tradjenta 5 mg once a day |
|
4 |
Eliquis 5 mg twice a day |
|
5 |
Entresto 97-103 mg twice a day |
|
6 |
torsemide 60 mg twice a day |
|
7 |
atorvastatin 40 mg once a day before bed |
|
8 |
Humalog 15 units as needed with meals |
|
9 |
Lantus 40-80 units once a day |
|
10 |
Vitamin D2 50,000 IU once a week |
|
11 |
Klor-Con M20 20 mEq twice a day |
|
- Based on the information provided in the case summary, it would appear that the patient did not have any intercurrent event which might explain his sudden change in status. That should probably be specifically stated in the case summary.
We have address this point at lines 273-282
- The authors provide laboratory information regarding renal function. They should include some information about troponin and BNP levels.
Supplementary table 2 lists BNP levels; however, we did not collect troponin data. For your reference, here are the BNP levels by date:
- 2428 pg/mL on 05/21/2020
- 2772 pg/mL on 05/22/2020
- 25767 pg/mL on 05/29/2020
- 16888 pg/mL on 06/02/2020
- 9709 pg/mL on 06/04/2020
- 5527 pg/mL on 06/11/2020
- The patient was placed on a plant-based diet which presumably helped his acid-base status. The authors should provide information about his initial weight at presentation and his weight at his final follow-up. It would also be helpful with the authors provided some basic information about this plant-based diet. For example, how many calories are provided in lipids, protein, and carbohydrates?
We agree that these details are important. We have body weight data indicated in supplementary table 2. Regarding the plant-based diet, we have provided additional details on this at lines 95-98
Reviewer 3 Report
Comments and Suggestions for Authors
Dear Authors,
I greatly appreciate the thoroughness and expertise demonstrated in your case description. The decision to discontinue unnecessary therapies, particularly the diuretics, and to employ hydration with 0.9% followed by 0.45% sodium chloride solution was both strategic and commendable. This approach not only highlighted the renal reserve function but also contributed to reactivating dormant nephrons, which ultimately played a crucial role in the patient's recovery. I fully support your manuscript for publication, with a few recommendations for further enhancement:
-
Explanation of Sodium Chloride Solutions: Please provide a brief explanation to the practical doctors for your choice of using 0.9% and 0.45% sodium chloride solutions. This would help clarify the rationale behind this treatment decision for clinicians who may not be as familiar with this approach.
-
Functional Renal Reserve: Since restoring functional renal reserve is a key point in your treatment strategy, I suggest you elaborate on how the sodium chloride solution plays a role in “triggering” this renal reserve mechanism.
-
Literature on Functional Renal Reserve: Please add relevant references on functional renal reserve, as this will provide a stronger evidence base and context for your treatment approach.
-
Clarification on Fluid Administration: In line 119, it is mentioned that the patient received 6 liters of IV fluids, but the table shows an infusion of 4 litres. Please clarify this discrepancy for accuracy and consistency in your data presentation.
Thank you for your attention to these suggestions, and I look forward to seeing your excellent work published.
Sincerely,
Author Response
I greatly appreciate the thoroughness and expertise demonstrated in your case description. The decision to discontinue unnecessary therapies, particularly the diuretics, and to employ hydration with 0.9% followed by 0.45% sodium chloride solution was both strategic and commendable. This approach not only highlighted the renal reserve function but also contributed to reactivating dormant nephrons, which ultimately played a crucial role in the patient's recovery. I fully support your manuscript for publication, with a few recommendations for further enhancement:
Thank you for the positive feedback.
Explanation of Sodium Chloride Solutions: Please provide a brief explanation to the practical doctors for your choice of using 0.9% and 0.45% sodium chloride solutions. This would help clarify the rationale behind this treatment decision for clinicians who may not be as familiar with this approach.
Functional Renal Reserve: Since restoring functional renal reserve is a key point in your treatment strategy, I suggest you elaborate on how the sodium chloride solution plays a role in “triggering” this renal reserve mechanism.
Literature on Functional Renal Reserve: Please add relevant references on functional renal reserve, as this will provide a stronger evidence base and context for your treatment approach.
We have addressed these points on lines 234-259
Clarification on Fluid Administration: In line 119, it is mentioned that the patient received 6 liters of IV fluids, but the table shows an infusion of 4 litres. Please clarify this discrepancy for accuracy and consistency in your data presentation.
The patient received 6 liters of intravenous fluids during his hospitalization (the first 2 liters were infused before the initial arterial blood gas measurement and are not shown in the graph). This is provided at lines 127-128.